# Protective Effects of Recombinant Human Angiogenin in Keratinocytes: New Insights on Oxidative Stress Response Mediated by RNases

**DOI:** 10.3390/ijms23158781

**Published:** 2022-08-07

**Authors:** Rosanna Culurciello, Andrea Bosso, Romualdo Troisi, Valentina Barrella, Ilaria Di Nardo, Margherita Borriello, Rosa Gaglione, Valeria Pistorio, Serena Aceto, Valeria Cafaro, Eugenio Notomista, Filomena Sica, Angela Arciello, Elio Pizzo

**Affiliations:** 1Department of Biology, University of Naples Federico II, 80126 Naples, Italy; 2Centro Servizi Metrologici e Tecnologici Avanzati (CeSMA), University of Naples Federico II, 80126 Naples, Italy; 3Department of Chemical Sciences, University of Naples Federico II, 80126 Naples, Italy; 4Department of Precision Medicine, University of Campania “Luigi Vanvitelli”, Via L. De Crecchio 7, 80138 Naples, Italy

**Keywords:** stress induced RNases, tiRNA, stress granules, skin cells, oxidative stress

## Abstract

Human angiogenin (ANG) is a 14-kDa ribonuclease involved in different pathophysiological processes including tumorigenesis, neuroprotection, inflammation, innate immunity, reproduction, the regeneration of damaged tissues and stress cell response, depending on its intracellular localization. Under physiological conditions, ANG moves to the cell nucleus where it enhances rRNA transcription; conversely, recent reports indicate that under stress conditions, ANG accumulates in the cytoplasmic compartment and modulates the production of tiRNAs, a novel class of small RNAs that contribute to the translational inhibition and recruitment of stress granules (SGs). To date, there is still limited and controversial experimental evidence relating to a hypothetical role of ANG in the epidermis, the outermost layer of human skin, which is continually exposed to external stressors. The present study collects compelling evidence that endogenous ANG is able to modify its subcellular localization on HaCaT cells, depending on different cellular stresses. Furthermore, the use of recombinant ANG allowed to determine as this special enzyme is effectively able to counter at various levels the alterations of cellular homeostasis in HaCaT cells, actually opening a new vision on the possible functions that this special enzyme can support also in the stress response of human skin.

## 1. Introduction

Cells are frequently subjected to different environmental stimuli that evoke acute or chronic stress. A network of cellular stress response programs helps cells to weather adverse conditions to renew cellular homeostasis [1]. These control programs act at two different levels by downregulating energy-intensive processes such as transcription and translation: a slow response, which provides for a transcriptional regulation that targets genes and leads to the production of proteins involved in stress adaptation and the repair of stress-induced damage [2]; a faster response, based on flexible changes of translational regulation, in which the response to stress occurs starting from pre-existing mRNA and proteins [3]. Although stress influences each step of the translation, the initiation phase is the major target of stress-induced translational silencing [4], consisting in the modification of components of the translational machinery. Translation is initiated when the preinitiation complex binds to capped mRNA in association with the eIF4E cap-binding protein, the eIF4A RNA helicase, and the eIF4G scaffold protein [5]. Several stress-activated signaling pathways inhibit protein synthesis by disabling this initiation program. General events in these pathways include: (1) phosphorylation of eIF2a; (2) phosphorylation of eIF4E-binding proteins; (3) phosphorylation of ribosomal protein S6; (4) inosine modification of double-stranded RNA; (5) endonucleolytic cleavage of ribosomal RNA. Combinations of these stress-activated inhibitory pathways play a major role in reprogramming protein translation in stressed cells [6,7,8,9,10]. Mechanisms of stress-induced protein synthesis inhibition also include the targeting of RNA components of the translational machinery—tRNAs in particular—by specific ribonucleases (RNases). RNases are ancient enzymes that catalyze the degradation of RNA into smaller fragments and regulate various aspects of RNA metabolism, including the cleavage of tRNA during a variety of stress responses [11]. Most RNases play key roles in the maturation, quality control and turnover of cellular RNA; other RNases are instead activated in response to stress conditions (stress-induced RNases-siRNases). In physiological conditions, siRNases are inactivated by several mechanisms including physical compartmentalization within membrane-bound organelles such as vacuoles or nuclei, secretion into the extracellular environment, or binding to RNase inhibitor [12]. Upon stress, these enzymes are activated, move into the cytoplasm, and cleave different RNA substrates that can profoundly affect cellular physiology [12]. Among the siRNases, particularly relevant is human RNase 5, also known as angiogenin (ANG), a member of the vertebrate RNase superfamily, widely studied for different physiological functions [13] and recently also characterized for its stress response role linked to tRNA cleavage. ANG, in contrast to most members of the vertebrate RNase superfamily, exhibits relatively low ribonuclease activity but is essential for its proliferative properties [14,15]. It is a 14 kDa cationic single-chain protein, characterized by three α-helices, seven β-strands and a 3_10_ helix (at the C-terminus), and it is reputed to be a protein secreted into the extracellular environment to stimulate the formation of new blood vessels and cell growth [15]. ANG was originally isolated from the conditioned medium of HT-29 human colon adenocarcinoma cells as a tumour angiogenic factor [16]; afterwards, many other additional roles were detected in a huge number of both physiological and pathological conditions [17]. Indeed, it is known that ANG is involved in cancer progression by stimulating both tumour angiogenesis and cancer cell proliferation. Moreover, it possesses neurogenic and neuroprotective activity [17]. In fact, ANG deficiency has been related to pathogenesis underlying neurodegenerative diseases including amyotrophic lateral sclerosis (ALS) [18] and Parkinson’s disease [19]. Several in vivo studies have shown that ANG inhibitors and activators are effective therapeutics for cancer [20] and ALS [21], respectively. Mechanistic studies also indicate that nuclear localization is necessary for ANG to exert the above indicated biological activities [22]. ANG is also involved in the innate immune system and inflammatory disease [23]. It has been shown that ANG, as a circulating protein induced during inflammation, exhibits microbicidal activity against bacterial and fungal systemic pathogens, suggesting that it can also contribute to systemic responses to infection [24]. ANG is also a paracrine agent that can be taken up by endocytosis into the cytoplasm of target cells [25,26]. Nevertheless, beyond the autocrine or paracrine origin, stress-induced ANG cellular activity is critically regulated by the endogenous inhibitor, RNH1, both in the nucleus and cytosol. In physiological growth conditions, ANG undergoes nuclear translocation and accumulates in the nucleolus where it stimulates rRNA transcription [12,13]. When cells are subjected to stress, ANG drastically changes its intracellular localization, mediating the cleavage of cytoplasmic mature tRNAs and leading to the production of stress-derived fragments [27]. These tRNA-derived stress-induced RNAs, also known as tiRNAs, contribute to the cell stress response and reveal potential cell protective effects, mainly via reprogramming translation, inhibiting apoptosis, degrading mRNA and contributing to stress granules (SGs) recruitment [28,29]. Although only recently identified, the stress-induced role of ANG, mainly focused on the release of tiRNAs, is now widely validated in numerous cell types [30,31,32]. However, to date, it is somewhat unique that there are no data to support the stress-induced role of ANG in skin, although it represents an effective barrier between the organism and the environment, providing the primary protection against several external dangers [33]. Starting from this consideration, the main objectives of this study were to verify the expression of ANG in skin cells and to assess the potential effects that this enzyme could exert when the homeostasis of these cells is upset by abiotic stresses.

## 2. Results

### 2.1. Expression and Localization of Endogenous ANG in HaCaT Cells

In order to evaluate the potential role of ANG in skin stress response, HaCaT cells were chosen as the experimental system. First, the basal level of expression of this protein in these cells was evaluated. As shown in Figure 1, analyses performed both by RT-qPCR and Western blot indicate that endogenous ANG is expressed in HaCaT cells, and its expression does not undergo substantial changes both at transcriptional and translational level when subjected to three different stress stimuli.

This early evidence would suggest that HaCaT cells constitutively synthesize ANG and that abiotic stresses do not alter its expression level. Therefore, to verify if the same abiotic stresses could alter the subcellular localization of ANG in HaCaT cells, immunofluorescence analyses were carried out. As shown in Figure 2A, in physiological conditions, ANG was localized both in cytosol and nuclei, in accordance with what is commonly reported for ANG in other cells [12]. Interestingly, when the cells were subjected to oxidative stress by sodium arsenite (SA) or H_2_O_2_ (Figure 2B,C), or to thermal stress (1 h at 45 °C) (Figure 2D), ANG dramatically changed its subcellular localization, clearly colocalizing with cytoplasmic loci, positive to immunostaining with Poly(A)-binding protein (PABP), a specific marker of SGs. This result, reported for the first time in skin cells, corroborates the shared idea that ANG is involved in the recruitment of SGs, and its potential stress-induced role is predominantly carried out in the cytosol.

To note, recent studies have also clearly established that ANG may represent a paracrine “help me” signal secreted from stressed cells that stimulates defense mechanisms in adjacent cells [19]. Furthermore, in different cell types it has been extensively verified that ANG can be taken up by sub-confluent cells from the extracellular matrix and translocate in the nucleus where it can contribute to ribosome biogenesis [15,34]. Taking all these premises together and considering that ANG properties have never been investigated on cells of the outermost layer of the skin, the next steps were exclusively based on exogenous proteins produced in recombinant form (rANGs), designed to better outline the contribution of this special enzyme in HaCaT cell stress response.

### 2.2. Effects of rANG on Viability and Homeostasis of HaCaT Cells

Exploiting a strategy that was already established in our research group, rANG was produced in *E. coli* cells (see Section 4) and purified to homogeneity [35]. Subsequently, before proceeding to verify its effects on HaCaT cells, all of the rANG variants taken into consideration in this paper were preliminary analysed to check their folding (Figure 3 and Figure 4), catalytic activity (Table 1) and biocompatibility (Figure 5).

The analyses performed by CD and fluorescence spectroscopies indicate that rANG shows a secondary structure and catalytic efficiency that are similar to those reported in previous studies [36,37]. Moreover, it does not show significant toxicity when administered to HaCaT cells at increasing doses up to 20 μM (Figure 5). After such confirmation, we proceeded to analyze the effects that pre-treatment with the exogenous protein could exert on HaCaT subjected to a stress stimulus promoted by SA. By an MTT test (Figure 6), we established that the stressful treatment still guaranteeing an acceptable and significant cell survival was 500 μM SA for 1 h. For this reason, all subsequent experiments were carried out using these conditions. Concerning the pre-treatment of cells with rANG, the conditions chosen in all the strategies (2 μM and 1 h at 37 °C) were established after a series of preliminary tests (not shown) that pointed out a significant impact of the protein on stressed cells without toxicity.

Firstly, rANG pre-treated HaCaT cells were analysed by an RT-qPCR, revealing an attenuated expression of the HSPA6 gene (encoding for stress-induced heat shock protein HSP70 [38]) in comparison with HaCaT cells that were subjected to the same perturbation in the absence of pre-treatment with the recombinant protein (Figure 7A). This protective trend was also confirmed in a further experimental approach in which intracellular ROS levels were analysed in HaCaT cells by a DCFH-DA assay (see Section 4). This test allowed us to detect increased ROS levels in stressed HaCaT cells that were not preliminarily incubated with rANG, whereas significantly reduced ROS levels were detected in HaCaT cells that were firstly treated with rANG and then stressed by SA, thus, further confirming the protective role exerted by rANG (Figure 7B).

To further investigate the rANG-mediated protective effect on HaCaT, a cytofluorimetric analysis with propidium iodide was also performed. The results clearly show that SA stress induces a strong accumulation of cells in G1 phase (+20%) compared to untreated cells. Meanwhile, this increase was markedly reduced in the stressed HaCaT cells that were pre-treated with rANG (Figure 7C). Concomitant with G1 arrest, a decrease in S/G2 phase was observed when HaCaT cells were exposed to SA. Interestingly, pre-treatment with rANG also prevented this alteration, further corroborating the involvement of rANG in the cellular stress response of HaCaT cells (Figure 7C).

### 2.3. Internalization of rANG_6xHis_ and Its Colocalization with SGs

The interesting behavior of rANG induced us to verify whether the enzyme can exert its properties once internalized in the HaCaT cells. To obtain this information, we designed a variant that should be discriminable from the endogenous ANG by the insertion in the C-term region of a 6xHis tag, useful to follow up its specific immunoreactivity. The variant, herein named rANG_6xHis_, includes a 7 aa-linker (KLAAALA) derived from the pET vector that works as a spacer between the protein and the 6xHis tag. A model of rANG_6xHis_ is shown Figure 8.

The recombinant variant was produced and purified to homogeneity, as described in Section 4. The folded state of rANG_6xHis_ was investigated in solution by CD spectroscopy. The inspection of the far-UV CD spectrum (Figure 3A) indicated that rANG_6xHis_ retained the secondary structure of the parent protein. Thermal unfolding curves were also obtained in the temperature range 20–85 °C by following the CD signal at 220 nm (Figure 3B). CD spectra recorded at 20 °C before heating and after cooling indicated the typical reversible thermal unfolding of RNases (Figure 4). The T_d_ of rANG was 59 °C, while that of rANG_6xHis_ was lower (55 °C), indicating a slight influence of 6xHis tag on the overall thermal stability of the protein. The catalytic efficiency of rANG_6xHis_ (Table 1) suggests that the modified protein still preserves the ribonucleolytic activity, although it was reduced with respect to that of the parent protein. Similar to rANG, prior to analyzing the rANG_6xHis_ properties on stress-induced cells, a preliminary analysis of its biocompatibility on HaCaT cells was carried out (Figure 9).

Once it was established that rANG_6xHis_ was correctly structured and non-toxic on HaCaT cells (up to 20 μM), its potential ability to be internalized by them was assessed by Western blot and immunofluorescence analyses (Figure 10). As shown in Figure 10A, the Western blotting profile of purified rANG_6xHis_ was well stackable with the bands corresponding to those detected in the lysate of the HaCaT cells that were administered with 2μM rANG_6xHis_ (Figure 10B), whereas no signals were detected in the control cells. Furthermore, in order to verify our ability to monitor the uptake of rANG_6xHis_ and to distinguish it from endogenous ANG, we tested an antibody specifically recognizing the His tag by a laser scanning confocal immunofluorescence analysis. The immunofluorescence analysis of HaCaT cells treated with 2μM rANG_6xHis_ for 1 h at 37 °C (Figure 10C) showed that the protein was internalized, localizing both in the cytoplasm and in the nucleus. This condition appeared not to compromise the cell homeostasis, since the PABP signal showed its canonical subcellular distribution. Moreover, the immunofluorescence analysis of HaCaT cells that were pre-treated with rANG_6xHis_ and then subjected to oxidative stress by SA (Figure 10D) showed a clear co-localization of the recombinant protein with SGs, thus, indicating that rANG_6xHis_, once internalized, is able to change its subcellular localization in relation to cell growth conditions.

It is interesting to note that rANG_6xHis_ also showed the same properties highlighted for rANG in stressed HaCaT cells, i.e., the ability to attenuate the HSPA6 expression, the level of ROS released, and to positively alter the cell cycle distribution of HaCaT cells under stress conditions (Figure 11A–C). All these data seem to indicate a close correlation between the stress-induced attitudes of ANG and its cytoplasmic localization.

### 2.4. Effects of H13A_rANG Variant on SA-Stressed HaCaT Cells

A further investigation was carried out to verify if it was possible to trace a connection between the above-mentioned stress-induced properties of ANG and its catalytic efficiency. To this purpose, the inactive H13A variant [39,40] was produced to homogeneity, as described in Section 4. A conformational analysis by CD spectroscopy revealed that H13A_rANG exhibits a similar secondary structure with respect to the parent angiogenin (Figure 3A) with a net reduction in the thermal stability (Figure 3B). It is well-known that in human angiogenin, H13 residue plays a crucial structural and functional role, and that its mutation induces a modification of the active site organization and a dislocation of Helix H1 of approximately 1.3 Å from its native position [41], thus, explaining the observed decrease in the thermal stability of the variant. Nonetheless, the global architecture of the protein remains essentially unchanged, thus, preserving the cell-binding site. Interestingly, as shown in Figure 12, the use of this catalytically compromised variant in HaCaT cells resulted in responses to stress that were not particularly incisive if compared to those that were obtained by analyzing native rANG-treated cells (Figure 3 and Figure 7). In detail, the expression of both the HSPA6 gene (Figure 12A) and ROS release (Figure 12B) appeared not to be markedly affected by the preventive presence of the H13A_rANG variant in SA-stressed HaCaT cells. Moreover, this “negative” trend was also confirmed by an analysis of the cell cycle (Figure 12C) in the SA-stressed HaCaT cells that were pre-treated with the H13A_rANG variant. Indeed, the collected data indicated that the alterations in G1 and S/G2 phases, observed in SA-stressed HaCaT cells, were not “positively” ameliorated as obtained in the stressed cells that were pre-treated with rANG and rANG_6xHis_, when the same pre-treatment was performed with H13A_rANG, underlining a possible correlation between the absence of a protective effect promoted by angiogenin and the peculiar properties of this variant, i.e., the lack of catalytic activity [39] and the low thermal stability.

## 3. Discussion

All mammalian cells exposed to adverse conditions respond to changing growth conditions by the regulation of gene expression and translational control. Although transcriptional adaptation is essential in calibrating the strength of stress response, regulation of global protein synthesis coupled with selective translation allows cells to maximize their survival under stress. Among the proteins that are particularly able to influence the translation rate in response to stress is certainly human angiogenin (ANG), probably the most popular stress-induced RNase studied in recent years [27,28,29,31]. Since its discovery [16], ANG and its various functional implications have been extensively studied in numerous cell types [30,31,32]. However, especially in relation to recent findings regarding its stress-induced properties, to the best of our knowledge, there was no accurate study of ANG in relation to human skin cells. This seemed surprising, above all because of the delicate role that these cells play as a protective barrier between the inside of the human body and the external environment. This unusual issue stimulated our interest in understanding if there is a concrete connection between ANG and skin cells and, to do this, we decided to investigate HaCaT cells under different growth conditions. Analyses performed on the endogenous protein (Figure 1 and Figure 2) revealed to us that ANG is localized both in the nucleus and in the cytosol. On the contrary, under different cell growth conditions, its synthesis does not undergo significant oscillations, while its localization becomes more cytosolic, overlapping in different points with the stress granules. To the best of our knowledge, such behavior has never been shown for ANG in skin cells and evokes what our group had already highlighted for different cell lines [12]: also in skin cells, ANG has a strategic subcellular location to exercise its stress-induced role, which is likely associated to its ability to hydrolyze tRNAs and release active tiRNAs. This observation is intimately linked to the central node around which this manuscript develops, i.e., to highlight whether a greater intake of ANG at the cytosolic level could alter the cellular response following perturbations of homeostasis. This issue was addressed by the use of recombinant ANG (rANG) to exploit the well-established concept that the protein is also able to act as a paracrine factor [25,26]. The collected results (Figure 7) clearly suggest that the pre-treatment of HaCaT cells with rANG causes a more incisive cellular response to a putative perturbation. In fact, in relation to this, the results suggest that rANG confers to HaCaT cells a better predisposition to counteract oxidative stress induced by sodium arsenite, both at the level of the selective expression of a heat shock protein (HSP70 [38]) and globally, in relation to ROS release and cell cycle regulation. Taken together, these data indicate that the preliminary administration of rANG to the cells confers to them, once stressed, the ability to re-shape at various levels the reaction to a perturbation. However, all these findings were attributable to a version of the protein that did not allow us to determine whether internalization was an essential requirement in modulating the cellular response to stress. For this reason, we designed and produced a variant of ANG with a His tag in the C-terminal region in order to make it distinguishable from endogenous ANG and, thus, to verify its putative intake by HaCaT cells. This variant (named rANG_6xHis_), endowed with regular folding and with a catalytic efficacy comparable to that of rANG (Figure 3 and Figure 4 and Table 1), induced in HaCaT cells a response to stress that was identical to that reported for the parental protein (Figure 11). However, the most interesting observations were the ability of HaCaT cells to internalize the protein and the propensity of ANG to modify its subcellular localization according to the growth conditions (Figure 10). To the best of our knowledge, no data in the literature report either the internalization of ANG in human skin cells or its propensity to colocalize with stress granules that are induced by sodium arsenite treatment and, in general, its ability to actively contribute to induce a positive reaction to stress. We strongly believe that this issue is significant, as in the future we might imagine a hypothetical topical use of ANG on skin cells that are particularly susceptible to certain stresses. The obtained data also highlight another interesting issue, i.e., the ability of both recombinant proteins (rANG and rANG_6xHis_) to influence the cellular response to the treatment with oxidizing agents. This is probably due to the identical folding of the two molecules (Figure 3) that might allow proteins to interact with specific targets, allowing them to wander in the cytosol during the stress response. Based on this hypothesis, we focused on the evaluation of the catalytic activity importance in the stress-induced role of ANG. The experimental design involved the heterologous production of the variant H13A_rANG [39,41], in which the catalytic site is significantly compromised due to the replacement of catalytic His13 with an Ala residue. We found that this His-Ala substitution altered the thermal stability of the purified protein without significantly affecting its secondary structure, if compared to the parental version of the protein (Figure 3). However, when the effects of this variant were analysed upon its administration to the cells subjected to stress, we found that this catalytically ineffective variant, while retaining a folding similar to that of the active protein, was not able to attenuate either the expression of HSPA6 or the release of ROS. Moreover, the effects on cell cycle were found to be negligible (Figure 12). Taken together, all these data appear to corroborate the idea that the catalytic activity of ANG, mainly directed at the release of specific tiRNAs, plays a leading role in its stress-induced properties. In summary, we strongly believe that the pioneering evidence collected in this work has the potential to launch a promising field of research on human skin and stress-induced RNases.

## 4. Materials and Methods

### 4.1. Heterologous Proteins Production and Purification

The recombinant production of human angiogenin and its variants was obtained by using an expression and purification protocol that was already described for other ribonucleases belonging to the same superfamily [42]. Briefly, the expression plasmids pET-22b(+) encoding rANG, rANG_6xHis_ or H13A_rANG were used to transform competent *E. coli* strain BL21(DE3) (Invitrogen^®^, San Diego, CA, USA). Cells were grown at 37 °C to an OD_600_ nm = 0.6, then induced with 0.4 mM of IPTG (isopropyl-1-thio-d-galactopyranoside) and grown overnight. Inclusion bodies were obtained after sonication and centrifugation and recombinantly expressed proteins were then purified by RP-HPLC [42,43].

### 4.2. Circular Dichroism Analysis

Circular dichroism (CD) measurements were carried out using a Jasco J-1500 spectropolarimeter equipped with a Peltier thermostatic cell holder. Far-UV measurements were carried out at a protein concentration of 0.1 mg ml^−1^ in 20 mM of MES/NaOH pH 6.0 and 100 mM NaCl at 20 °C, using a cell with an optical path length of 0.1 cm. Spectra, registered with a 50 nm min^−1^ scanning speed, 2 s D.I.T., 1 nm data pitch, and 2.0 nm bandwidth, were obtained, averaging three scans. Thermal unfolding curves were obtained by following the CD signal at 220 nm in the 20–85 °C range at a heating rate of 30 °C h^−1^. The denaturation temperatures (T_d_) were determined through an analysis of the first derivative of the melting profiles. The reversibility of the transition was checked by lowering the temperature to 20 °C and re-collecting the spectrum.

### 4.3. Ribonucleolytic Activity Assays

Ribonucleolytic activity studies were performed according to reported procedures [42,44], with minor modifications. RNase A was purchased from Merck KGaA (Darmstadt, Germany) and the fluorogenic substrate 6-FAM-(dA)_2_rU(dA)_3_-6-TAMRA (where 6-FAM refers to 6-carboxyfluorescein and 6-TAMRA refers 6-carboxytetramethylrhodamine) was synthesized by Eurofins genomics.

Fluorescence measurements were performed at 20 °C under magnetic stirring in a quartz cuvette with a 1-cm optical path length, using a Fluoromax-4 spectrofluorometer equipped with a Peltier temperature controller. The variation of the emission signal induced by substrate cleavage was monitored at λ_em_ = 515 nm (slit width = 3 nm) using an excitation wavelength λ_ex_ = 490 nm (slit width = 2 nm). Each measurement was performed in triplicate in 100 mM of MES/NaOH pH 6.0 and 100 mM NaCl, using 100 nM 6-FAM-(dA)_2_rU(dA)_3_-6-TAMRA; 200 pM RNase A; or 500 nM rANG or rANG_6xHis_ to obtain a stable signal in the absence of the enzyme. Fluorescence was preliminarily recorded during an equilibration period of 5 min.

The catalytic reactions were initialized upon the addition of the enzymes and then monitored for ~15 min, after which time RNase A was added to bring the reaction to completion. The low ribonucleolytic activity of angiogenin does not allow for full cleavage within a reasonable time. The variation of the emission intensity with time was used to calculate the k_cat_/K_M_ value by the following equation:(1)$$\frac{k_{\mathrm{cat}}}{K_{M}}=\left(\frac{\frac{\Delta F}{\Delta t}}{F_{\max}-F_{0}}\right)\;\frac{1}{\left[E\right]}$$
where ΔF/Δt is the slope of the straight line that fits the initial velocity of the enzymatic activity with R^2^ ≥ 0.99, F_max_ and F_0_ are the fluorescence acquired after the complete cleavage of the substrate and in the absence of enzyme, respectively, and [E] is the concentration of the enzyme used during the experiment.

### 4.4. Cell Culture and Treatments

HaCaT cells were cultured in Dulbecco’s Modified Eagle Medium (DMEM) supplemented with 10% fetal bovine serum (FBS), 1mM of L-glutamine and 1% *v*/*v* penicillin/streptomycin solution (100 Unit/mL), and grown at 37 °C in 5% CO_2_.

Oxidative stress was induced with 500 µM SA (Merck KGaA, Darmstadt, Germany) for 1 h or 1 mM hydrogen peroxide (Merck KGaA, Darmstadt, Germany) for 2 h [45]; thermal stress was obtained exposing cells for 1 h to 45 °C in 5% CO_2_.

Recombinant proteins were administrated for the biocompatibility assay at increasing concentrations for 24 h. For the following experiments, pre-treatments were carried out by using 2 µM rANG, rANG_6xHis_ or H13A_rANG, for 1 h.

### 4.5. Cellular Extract

The cells were collected in ice cold phosphate buffered saline (PBS) and lysed in RIPA Buffer (50 mM Tris-HCl buffer at pH 8 containing 150 mM of NaCl, 1 mM of EDTA, 0.1% SDS, 1% Triton X-100 and protease inhibitors 1X) for 30 min in ice. After centrifugation at 14,000 rpm for 10 min at 4 °C, supernatants were collected and analyzed. Protein concentration was measured by the Bradford method (Merck KGaA, Darmstadt, Germany).

For the uptake analysis of rANG_6xHis_ by HaCaT cells, the resulting lysates of control and 2 µM rANG_6xHis_ treated cells were loaded on HisPur™ Ni-NTA Resin (Thermo Scientific^TM^, Boston, MA, USA) and an IMAC chromatography in denaturing conditions was performed, according to the manufacturer instructions. Elution fractions were collected and analyzed by Western blotting.

### 4.6. Western Blot Analysis

Proteins from total cellular lysis or from IMAC elution fractions were separated by SDS-PAGE, and then electro-transferred to Immobilon-PVDF membranes (Millipore). After blocking, PVDF membranes were incubated overnight in primary antibodies: rabbit angiogenin (ANG) polyclonal antibody (OriGene Technologies Inc., Rockville, Maryland, USA, 1:500); rabbit β-actin polyclonal antibody (Merck KGaA, Darmstadt, Germany, 1:1000); mouse 6x-His Tag mAb (Invitrogen^TM^). After washing, membranes were incubated with horseradish peroxidase (HRP) conjugated to goat anti-mouse or anti-rabbit IgG (1:10,000) (ImmunoReagents Inc., Chapel Hill Road, Suite 153, Raleigh, NC, USA). The membranes were then incubated with ECL chemiluminescent substrate and recorded by ChemiDoc Imaging System (BioRad, Segrate, Milano, Italy). The results were then subjected to a densitometric analysis.

### 4.7. Densitometric Analysis

A densitometric analysis was performed with the software ImageJ. To reduce the signal to-background ratio, the area which determines the signal intensity was also measured for the background, which was near to the signal of interest. Appropriate background intensity was subtracted from each signal intensity. Each analysis was performed in triplicate.

### 4.8. Cell Viability Assay

For the biocompatibility evaluation of rANG, rANG_6xHis_ or H13A_rANG, 5 × 10^3^ cells were seeded in a 96-well plate and, after 24 h of growth, treated with increasing amounts of recombinant proteins, or SA, as mentioned above.

The cells’ viability was then assessed by the MTT method [46] and analyzed at 570 nm by means of a Multi-Mode Microplate Reader (Synergy™ H4).

### 4.9. DCFH-DA Assay

The ROS quantification assay was carried out by using the DCFH-DA (2′,7′-Dichlorofluorescin diacetate) [47].

2 × 10^4^ cells were seeded into a 96-well plate and incubated at 37 °C in a 5% CO_2_ atmosphere overnight. Then, the cells were washed in PBS 1X, opportunely treated, and incubated with 20 µM of DCFH-DA at 37 °C for 40 min. After the incubation time the fluorescence of the cells from each well was measured and recorded in the excitation/emission wavelengths of 485–532 nm, by means a Multi-Mode Microplate Reader (Synergy™ H4).

### 4.10. Real-Time Quantitative PCR (RT-qPCR)

Gene expression was evaluated in HaCaT cells by an RT-qPCR. In brief, 3 × 10^5^ cells were plated and, after 24 h, exposed to treatments (see Section 4.4). The total RNA was then isolated using TRIzol™ Reagent (Invitrogen™) according to the manufacturer’s instructions, and 1000 ng of RNA was used for cDNA synthesis using SuperScript™ IV VILO™ Master Mix (Invitrogen™) following the manufacturer’s protocol. An RT-qPCR was performed by using SYBR GREEN PCR MASTER MIX (InvitrogenTM) in the StepOnePlus™ Real-Time PCR System (Applied Biosystems™). The expression of each gene detected was normalized with respect to GAPDH gene (housekeeping gene control). (see Table 2).

### 4.11. Immunofluorescence

Immunofluorescence analyses were performed on 2.5 × 10^4^ HaCaT cells that were seeded on coverslips and cultured for 24 h. After the treatments (see Section 4.4), the cells were fixed in 4% paraformaldehyde (PFA), permeabilized in 0.1% TRITON-X100 and blocked with 1% BSA. Then, the cells were incubated overnight with primary antibodies suitably diluted, as indicated by the manufacturer, in BSA 1%. The primary antibodies used were mouse 26-2F Angiogenin mAb (eBioscience™, Invitrogen^TM^); mouse 6x-His Tag mAb (Invitrogen^TM^); and rabbit PABPC1 pAb (Merck KGaA, Darmstadt, Germany). After washing in PBS, the coverslips were incubated with secondary antibodiesAlexa488 conjugated goat anti-rabbit (Invitrogen^TM^) or Cy3 conjugated goat anti-mouse F(ab’)2 (1:500 dilution) (Jackson ImmunoResearch Laboratories, West Grove, PA, USA) for 1 h. The nuclei were stained with DAPI (Molecular Probes, Invitrogen, Italy). After washing, the coverslips were mounted in Mowiol^®^ 4-88. Images were acquired using Zeiss Confocal Microscope LSM 700 at 63× magnification.

### 4.12. Cell Cycle Analysis

A cell cycle analysis was carried out by exploiting propidium iodide staining. After the treatments, 2.5 × 10^5^ cells were collected and resuspended in 500 μL of hypotonic buffer (0.1% Triton X-100, 0.1% sodium citrate, 50 μg/mL of iodide propidium, RNase A). The cells were then incubated in the dark for 30 min and samples were acquired on a FACS-Calibur flow cytometer using the Cell Quest software (Becton Dickinson, Milano, Italy) and ModFitLT version 3 software (Verity, Topsham, ME, USA).

### 4.13. Statistical Analysis

Statistical analyses were carried out by using GraphPad Prism. Values are reported as the means ± SEM of biological replicates (* *p* < 0.05, ** *p* < 0.01, *** *p* < 0.001 or **** *p* < 0.0001) compared to the respective controls (one-way ANOVA, followed by Bonferroni’s post-test).

## Figures and Tables

**Figure 1 ijms-23-08781-f001:**
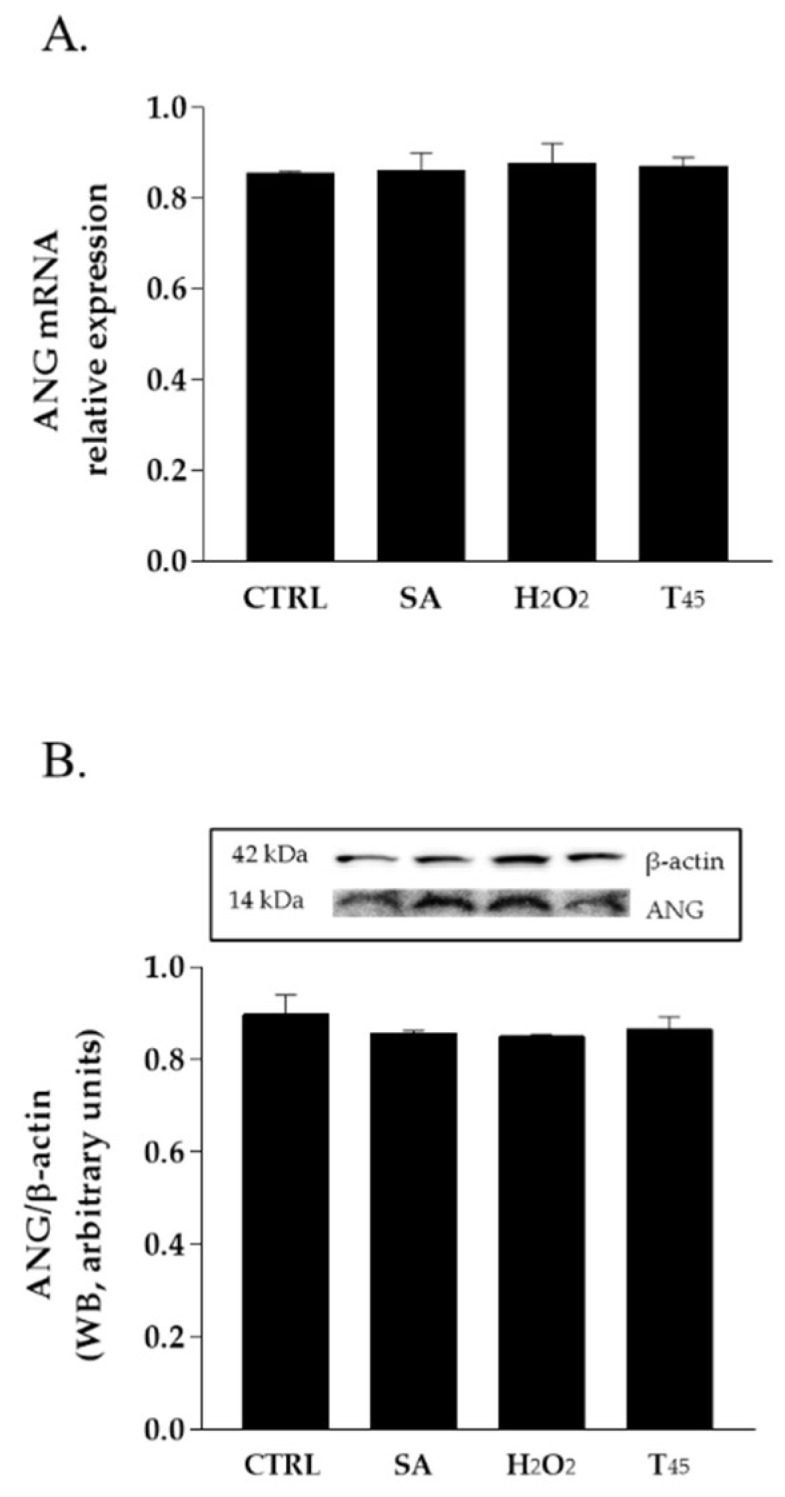
Analysis of ANG expression level in HaCaT cells cultured in different growth conditions. (**A**). Expression analysis of ANG mRNA by RT-qPCR; (**B**). Western blot analysis of cell lysates. CTRL: cells cultured in physiological conditions; SA: HaCaT cells treated with 500 µM sodium arsenite (SA) for 1 h at 37 °C; H_2_O_2_: HaCaT cells treated with 1 mM of hydrogen peroxide for 2 h at 37 °C; T_45_: HaCaT cells subjected to thermal stress at 45 °C for 1 h.

**Figure 2 ijms-23-08781-f002:**
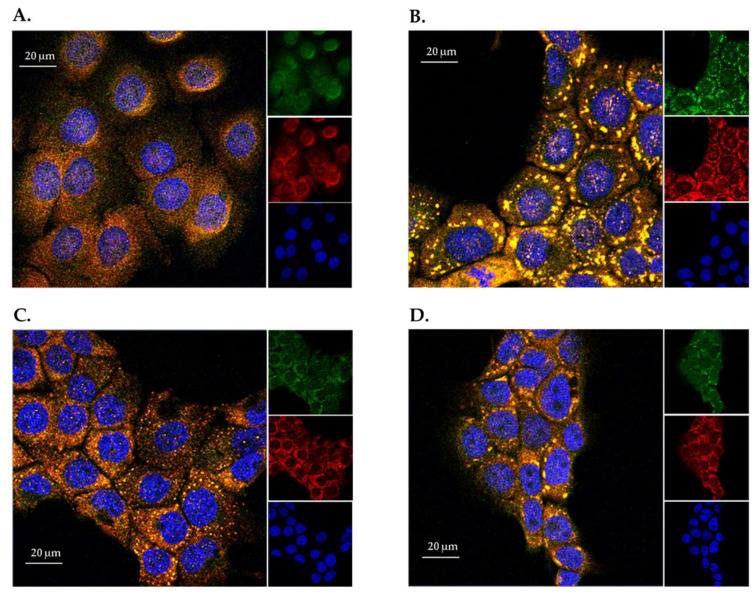
Immunofluorescence analyses of ANG sub-cellular localization under physiological and stresses conditions. (**A**) HaCaT cells cultured in physiological conditions; (**B**) HaCaT cells treated with SA for 1 h at 37 °C; (**C**) HaCaT cells treated with 1 mM of hydrogen peroxide for 2 h at 37 °C; (**D**) HaCaT cells subjected to thermal stress at 45 °C for 1 h. Cells were co-stained with PABP (red), ANG (green) and blue staining (DAPI) for nuclei. Yellow spots indicate the co-localization between ANG and PABP at the level of SGs.

**Figure 3 ijms-23-08781-f003:**
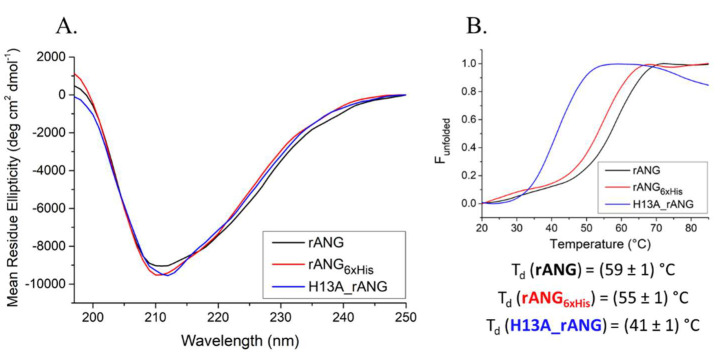
CD analysis of rANG and its variants. (**A**) CD spectra recorded at 20 °C in 20 mM of MES/NaOH pH 6.0 and 100 mM NaCl, using a protein concentration of 0.1 mg mL^−1^; (**B**) thermal denaturation profiles, as followed by monitoring changes in the molar ellipticity at 220 nm as a function of temperature. The denaturation temperature (T_d_) for each protein is shown in panel B.

**Figure 4 ijms-23-08781-f004:**
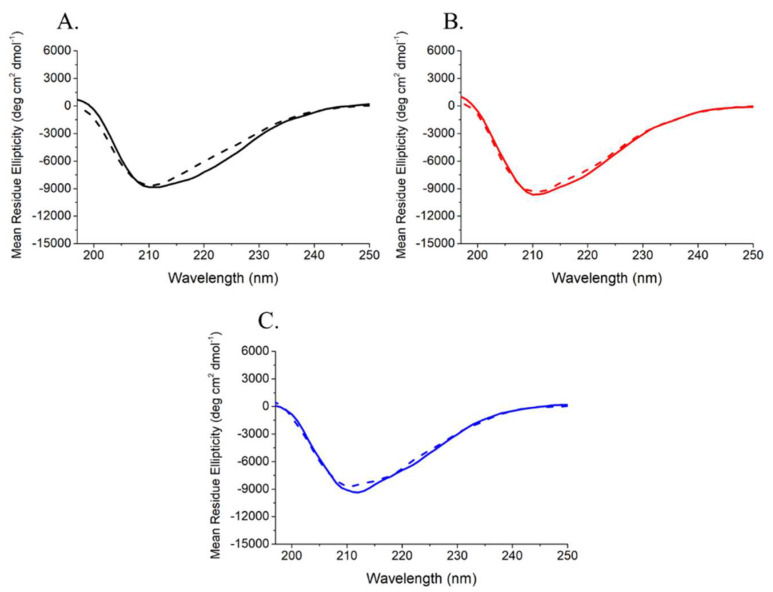
Superimposition of the CD spectra of (**A**) rANG, (**B**) rANG_6xHis_ and (**C**) H13A_rANG, registered at 20 °C before heating (plane line) and after cooling (dashed line). Spectra were recorded in 20 mM of MES/NaOH pH 6.0 and 100 mM NaCl, using a protein concentration of 0.1 mg mL^−1^.

**Figure 5 ijms-23-08781-f005:**
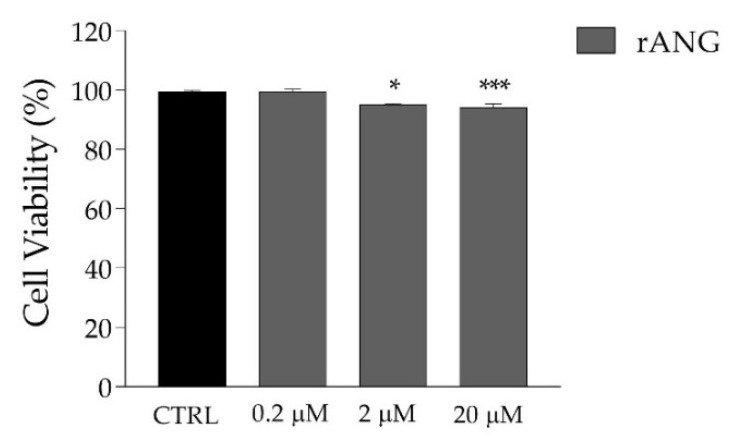
HaCaT cells viability measured after the treatment of cells for 24 h with increasing doses of rANG. Statistical analyses were carried out by using GraphPad Prism. Values are the means ± SEM of biological replicates (* *p* < 0.05, *** *p* < 0.001) compared to the respective controls (one-way ANOVA, followed by Bonferroni’s post-test).

**Figure 6 ijms-23-08781-f006:**
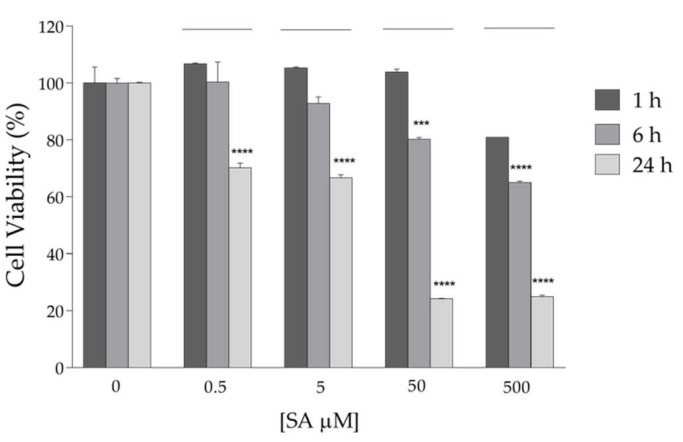
HaCaT cells viability measured after the treatment of cells with increasing concentration of SA at three different incubation time (1, 6 and 24 h). Experiments were performed in triplicate and statistical analysis were carried out by using GraphPad Prism. Values are the means ± SEM of biological replicates (*** *p* < 0.001 or **** *p* < 0.0001) compared to the respective controls (one-way ANOVA, followed by Bonferroni’s post-test).

**Figure 7 ijms-23-08781-f007:**
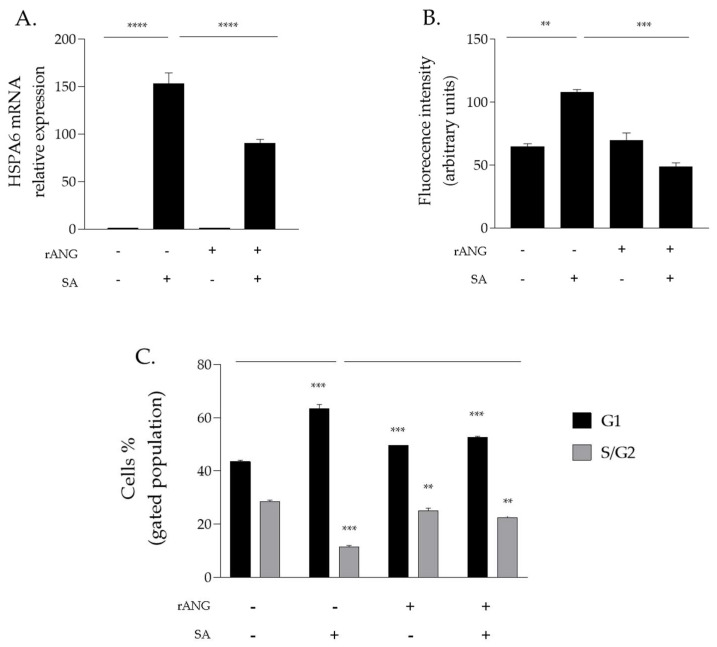
rANG effects on stress response of HaCaT cells. (**A**) RT-qPCR analysis of the expression of HSPA6 gene; (**B**) intracellular ROS detection by DCFH-Da assay; (**C**) distribution of HaCaT cells population among cell cycle G1 phase and S/G2 phase. (−) untreated cells; (+) pre-treatment of cells with 2µM rANG or 500 µM SA. Values are the means ± SEM of biological replicates (** *p* < 0.01, *** *p* < 0.001, **** *p* < 0.0001) compared to the respective controls (one-way ANOVA, followed by Bonferroni’s post-test).

**Figure 8 ijms-23-08781-f008:**
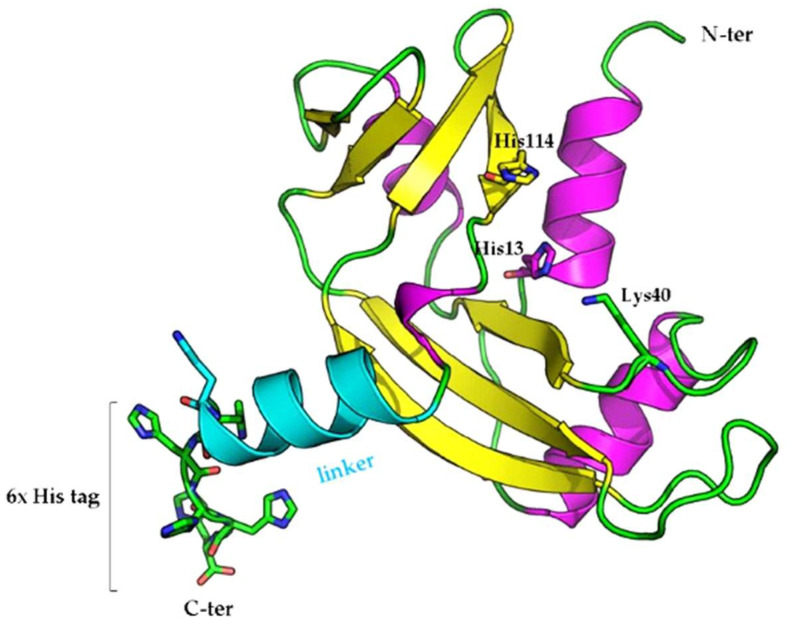
Model of the variant rANG_6xHis_. The linker peptide is highlighted in turquoise. Amino acids from the catalytic triad and the 6xHis tag are shown as sticks. The image was generated by the molecular visualization software PyMOL (https://pymol.org, accessed on 10 May 2022).

**Figure 9 ijms-23-08781-f009:**
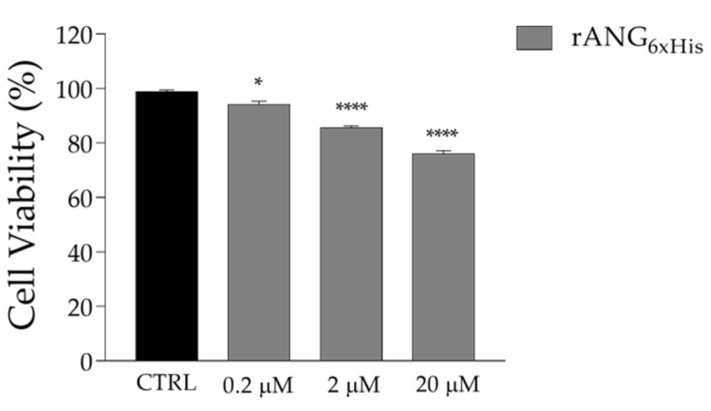
HaCaT cells viability measured after the treatment of cells for 24 h with increasing doses of rANG_6xHis_. Statistical analyses were carried out by using GraphPad Prism. Values are the means ± SEM of biological replicates (* *p* < 0.05, **** *p* < 0.0001) compared to the respective controls (one-way ANOVA, followed by Bonferroni’s post-test).

**Figure 10 ijms-23-08781-f010:**
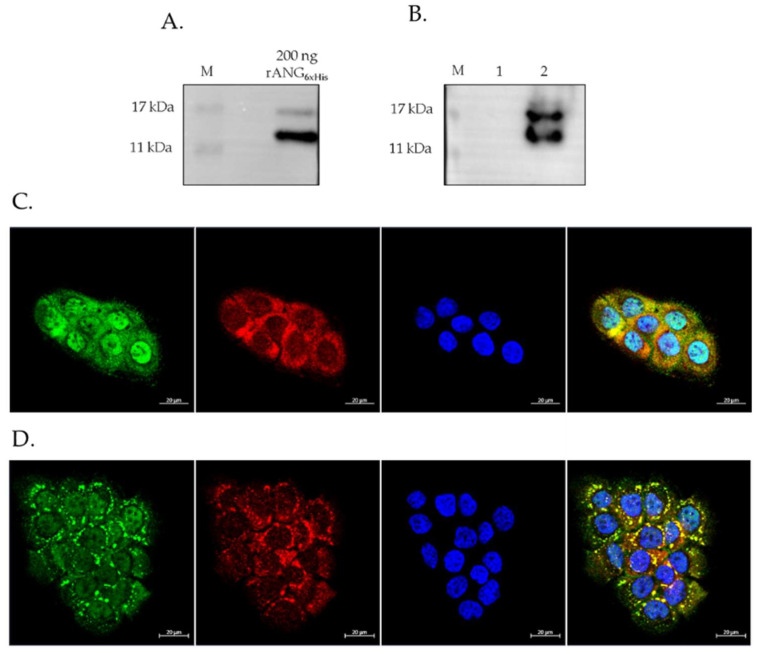
rANG_6xHis_ internalization and localization in HaCaT cells. **(A**) Western blot analysis of 200 ng of purified rANG_6xHis_ used for cells treatments; (**B**) Western blot analysis of cell lysates obtained by IMAC chromatography (M: protein ladder; 1: untreated HaCaT cell lysate; 2: 2 µM rANG_6xHis_ treated HaCaT cell lysate). (**C**) Immunofluorescence analysis of unstressed 2 µM rANG_6xHis_ pre-treated cells. (**D**) Immunofluorescence analysis of SA-stressed 2 µM rANG_6xHis_ pre-treated cells. PABP (red), rANG_6xHis_ (green) and blue staining (DAPI) for nuclei.

**Figure 11 ijms-23-08781-f011:**
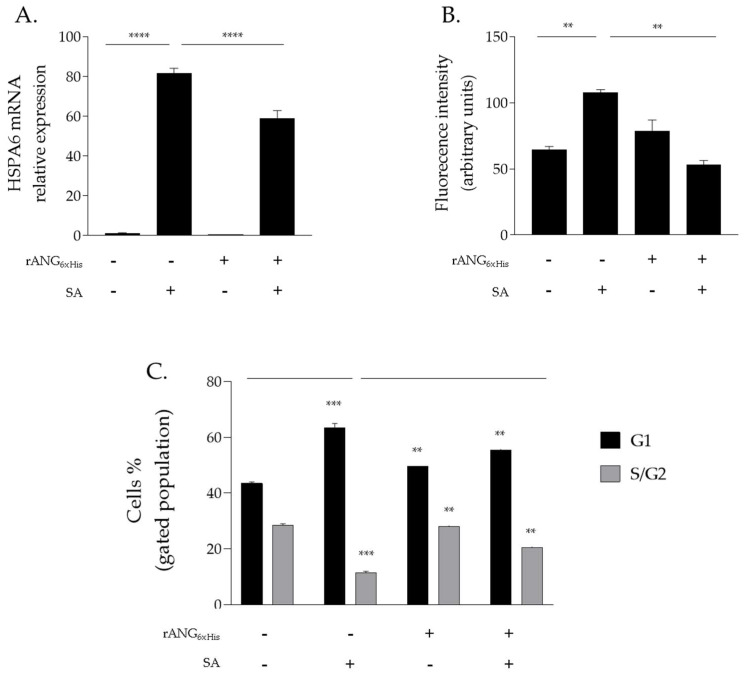
rANG_6xHis_ effects on HaCaT cells stress response. (**A**) RT-qPCR analysis of the expression of HSPA6 gene; (**B**) intracellular ROS detection by DCFH-Da assay; (**C**) distribution of HaCaT cells population among cell cycle G1 phase and S/G2 phase. (−) untreated cells; (+) pre-treatment of cells with 2 µM rANG_6xHis_ or 500 µM SA. Values are the means ± SEM of biological replicates (** *p* < 0.01, *** *p* < 0.001, **** *p* < 0.0001) compared to the respective controls (one-way ANOVA, followed by Bonferroni’s post-test).

**Figure 12 ijms-23-08781-f012:**
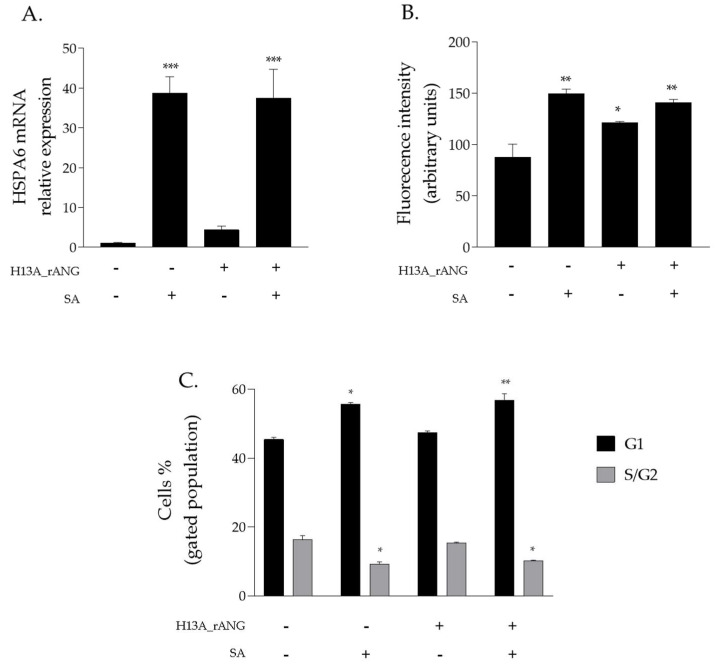
H13A_rANG effects on HaCaT cells stress response. (**A**) Real-time qPCR analysis of the expression of HSPA6 gene; (**B**) intracellular ROS detection by DCFH-Da assay; (**C**) distribution of HaCaT cells population among cell cycle G1 phase and S/G2 phase. (−) untreated cells; (+) pre-treatment of cells with 2 µM H13A_rANG or 500 µM SA. Values are the means ± SEM of biological replicates (* *p* < 0.05, ** *p* < 0.01, *** *p* < 0.001) compared to the respective controls (one-way ANOVA, followed by Bonferroni’s post-test).

**Table 1 ijms-23-08781-t001:** Ribonucleolytic activity of rANG and rANG_6xHis_ compared with that of RNase A. Each value is calculated as the average of three measurements.

	k_cat_/K_M_ (M^−1^ s^−1^)
RNase A	(4.6 ± 0.5) × 10^7^
rANG	(5.2 ± 0.3) × 10^3^
rANG_6xHis_	(1.6 ± 0.2) × 10^2^

**Table 2 ijms-23-08781-t002:** Primer sequences associated to selected genes.

Genes	Forward	Reverse
GAPDH	5′-CACCACACTGAATCTCCCCT-3′	5′-TGGTTGAGCACAGGGTACTT-3′
ANG	5′-CACTTCCTGACCCAGCACTA-3′	5′-ATGTCTTTGCAGGGTGAGGT-3′
HSPA6	5′-TGCAAGAGGAAAGCCTTAGGGACA-3′	5′-TTTGCTCCAGCTCCCTCT TCTGAT-3′

## Data Availability

Not applicable.

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
