# Peer review of "Protective Effects of Recombinant Human Angiogenin in Keratinocytes: New Insights on Oxidative Stress Response Mediated by RNases"

_ijms, 2022, doi:10.3390/ijms23158781_

Round 1
Reviewer 1 Report
Angiogenin, also known as ribonuclease 5, was originally identified as a tumor angiogenic factor, but its biological activity has been extended from inducing angiogenesis to stimulating cell proliferation and to promoting cell survival. Due to the above properties, the use of angiogenin for therapy is being actively studied. Thus, it has been shown that it is effective for accelerating the healing of long-term healing wounds in diabetes mellitus, the use of its activators in sclerosis, and inhibitors in oncological pathology. This work demonstrated that both endogenous and exogenous angiogenin are involved in cell protection under stressful conditions. Methods fit the task The article analyzes in detail the molecular mechanisms of the influence of synthesized angiogenins. Methods fit the task The text is well structured, written in good scientific language. However, there are some comments and questions.
1. It is not clear why in the text of the main article the figures are numbered Figure S1, Table S1.
2. Not clear sentence: «Similarly to rANG, also for rANG6xHis preliminary biocompatibility tests were carried out prior to the assays on cells». What preliminary tests of biocompatibility to cells is in question, if the results obtained on cells are presented.
3. The authors also claim that it is not toxic, but the viability was checked after 24 hours, the cells decreased by 20%. Perhaps when cultivating for 3 days, the effect was seen more clearly.
4. The authors mention that angiogenin stimulates cell proliferation. However, this fact is not confirmed either in the references or in the data. On the contrary, with an increase in the concentration of Angiogenin, cell proliferation is inhibited. How do the authors explain this fact? Pay close attention to links. So in ref. 14 the Ribonucleolytic Activity of Angiogenin is described in detail, but nothing is said about proliferation, however, the authors present this article as a demonstration of the effect of angiogenin on cell proliferation.
5. It is not entirely clear why both sodium arsenite (SA) and H2O2 are used to model oxidative stress. Different mechanisms of protection against them? Moreover, some of the results are shown only with a SA. Also the concentration 1 mM hydrogen peroxide is very high for cells. Figure 2 shows that the cells are reduced in size, significantly stressed, and vacuoles are formed.
6. «To further investigate the rANG-mediated protective effect on HaCaT, a cytofluorimetric analysis was also performed.» It is not entirely clear the sentence, cytofluorimetric analysis of what? In this case, the authors use the method to analyze the cell cycle, which should be reflected. I would also suggest redoing Figures 4, 6 and 7. Combine all phases of the cell cycle into one chart.
7. How do you explain the more significant decrease in ROS concentration when exposed to both angiogenin and oxidative stress, compared with only angiogenin and control (Fig. 4 B).
Author Response
We thank the referee for his/her comments and suggestions that allowed us to improve our work. Our detailed comments and answers are attached in a separate file.

Reviewer 2 Report
The major finding of the manuscript is that ANG plays protective role under various stress in HaCat cells while catalytically inactive ANG is not able to exert its protective function.
However, there are following major concerns that needs to be addressed before accepting this manuscript.
1) The major stress for skin is UV rays as it is exposed to sunlight. It is more relevant to use UV irradiation as a source of stress on HaCat cells and check if ANG is protective in UV stress.
2) Catalytically inactive ANG (H13A_rANG) is proposed to lack protective function. However, it is possible that this modified ANG variant lack ability to internalize in the cell or subcellular localization. Thus, whether catalytic inactivity or inability to subcellular localization is responsible for lack of protective effect of mutant ANG? is not clear.
No data has been provided to confirm the internalization of H13A_rANG, its subcellular localization in homeostatic condition and after stress. Recombinant H13A_rANG with hist tag should be used to confirm this or ANG knockout cell-line should be generated and treated with recombinant H13A_rANG. Subsequently, internalization and subcellular localization should be studied.
3) Does HaCat cells possess receptor for ANG. qPCR or WB or RNA expression in publicly available RNA-seq data should be provided ideally for various skin cell-types.
4) ANG is a secretory protein and its secretion outside cells in stress can have propounding effect on neighboring cells. In fig1 author claim that ANG levels not change after various stresses in HaCat cells. Author should confirm if it gets secreted outside cell after these stresses.
5) Since cytoplasmic ANG is shown to generate tiRNAs, authors should show levels of tiRNAs in H13A_rANG in comparison to rANG6xHis under homeostatic condition and after stress in ANG knockout HaCat cells. Authors should also discuss how tiRNA production contributes to protection from stress.
Minor comments:
1) Unedited original western blot images should be provided to reviewers for authenticity.
2) Manuscript have multiple spelling mistakes e.g “no-preated” (line 212), “rybonucleotyc” (line 254), “althought” (line 255) etc., grammatical errors and some sentences are beyond the levels of understanding. Thorough check and rewriting are needed.
Author Response

(The authors gave the same response as above.)

Round 2
Author Response
We thank the referee for his/her comments and suggestions. Our detailed answers are in blue in the attached file.

Round 3
Reviewer 2 Report
Authors sufficiently addressed the comments.